# In-conduit capture of sub-micron volcanic ash particles via turbophoresis and sintering

Jamie I. Farquharson [1] ✉, Hugh Tuffen [2], Fabian B. Wadsworth[3], Jonathan M. Castro [4], Holly Unwin[2] & C. Ian Schipper[5]

Ash emission in explosive silicic eruptions can have widespread impacts for human health, agriculture, infrastructure, and aviation. Estimates of the total grainsize distribution (TGSD) generated during explosive magma fragmentation underpins eruption models and ash dispersal forecasts. Conventionally, the TGSD constrained via erupted deposits is assumed to match the TGSD produced at explosive fragmentation. Here we present observations from within the vent of a recent rhyolitic eruption (Cordón Caulle, Chile, 2011–2012), demonstrating that fine (<63 μm diameter) and ultra-fine (<2.5 μm diameter) ash particles are captured and sintered to fracture surfaces, and thus sequestered in the shallow subsurface, rather than emitted. We establish a conceptual model—uniquely contextualised through a combination of syn-eruptive observations and detailed post-eruption field investigation—in which turbophoresis (particle migration towards zones of lower turbulence) and rapid sintering create an inverse relationship between particle size and the probability of its subsurface capture. Such size-dependent capture efficiency preferentially removes submicron-diameter ash from the erupted componentry, decoupling the erupted size distribution from magmatic source conditions and potentially playing an important role in modulating eruption dynamics.

Understanding the controls on silicic volcanic eruption style is a grand challenge in geoscience[1]. Silicic eruptions—and rhyolitic eruptions in particular—remain enigmatic, in part because very few such eruptions have been observed. While key observations of the preserved and variably eroded deposits of past eruptions have proved invaluable in our understanding of silicic eruptive dynamics[2–5], recent eruptions at Chaitén (2008–2009) and Cordón Caulle (2011–2012), both in Chile, were witnessed directly and have allowed us to observe how rhyolitic eruptive processes proceed in real-time[6–9]. The blurred transition between explosive eruptions and subsequent lava effusion, together with evidence for shallow in-conduit reassembly and welding of pyroclastic debris, has led to a model in which effusive lava may be produced top-down from the products of the explosive phase[2,10]. This is consistent with the presence of sintering textures in all phases of a silicic eruption: obsidian pyroclasts and welded pumice in fall deposits[11–13], tuffisites in volcanic bombs[14–16], and partially sintered surfaces in the effusing lava itself[2]. This model raises questions about the poorly constrained structure of the uppermost conduit feeding the eruption as well as the relationship between emitted products and the at-depth source parameters.

Key to determining eruptive source parameters is the total grainsize distribution (TGSD) of erupted volcanic ash particles, which controls their subsequent transport and sedimentation[17,18], related respiratory health hazards[19], their preservation as geochronologically

[1]Institut Terre et Environnement de Strasbourg, UMR 7063, Université de Strasbourg, Strasbourg, France. [2]Lancaster Environment Centre, Lancaster University, Lancaster LA1 4YQ, UK. [3]Department of Earth Sciences, Durham University, Durham DH1 3LE, UK. [4]Institute of Geosciences, Johannes Gutenberg Universität, Mainz D-55128, Germany. [5]School of Geography, Environment and Earth Sciences, Victoria University of Wellington, Wellington 6012, New Zealand. ✉e-mail: jifarq89@googlemail.com

important tephra layers[20], their geochemical reactivity[21], and their use as indicators of the magnitudes of past volcanic events[22–24]. An important question remains: how does the TGSD of particles generated via subsurface fragmentation relate to the size distribution of particles emitted into the atmosphere and fed into plumes? Dispersal models assume an inverse relationship between particle size and mobility, with the smallest ash fraction found most distal to the vent due to its low settling velocity[25], and larger clasts being constrained closer to the vent[26]. However, recent field observations demonstrate the existence of 0.1–100 μm variably sintered ash particles coating the fractures that provided pathways for ash emission[2]. The demonstrable capture of fine ash within the vent itself shows that a re-evaluation of microscale ash emission processes is necessary. In this study, we present microtextural data from within the Cordón Caulle vent, and build a conceptual model for the capture of an ultra-fine ash fraction within the shallow vent architecture. We further highlight implications of the fact that the emitted products of explosive fragmentation may be fundamentally decoupled from the eruption source parameters during silicic eruptions.

## Results and discussion

### In-vent observations from the 2011–2012 Cordón Caulle eruption

The 2011–2012 Cordón Caulle eruption (Fig. 1a) was initially characterised by a Plinian phase during which the ash plume reached >15 km height (Fig. 1b); this soon gave way to a ten-month-long hybrid explosive–effusive phase characterised by intermittent ash plumes, with synchronous lava effusion and explosive activity from a common vent[27]. Syn-eruption videography in January 2012 revealed near-continuous ash venting with sporadic Vulcanian blasts occurring from two highly-fractured ~50–80 m-diameter lava piles within a single coalesced, horseshoe-shaped tephra cone[9]. Ash plumes, 2–5 km high, emanated from clusters of sub-vents ranging from near-point sources to arcuate, 10–30 m-long fractures (Fig. 1c–e). Sub-vents exhibited variable transitions between quiescence, steam venting, dilute ash venting, and incandescent bomb emission over timescales of 1–10 s, with apparent correlations between separate sub-vent activity indicating their subsurface connectivity. Bombs exited the vent at ~100 m s⁻¹, and largely fell within the tephra cone or on raised ground to the north and north east.

Field investigation in January 2014, shortly following cessation of eruptive activity, allowed the correlation of the syn-eruptive observations of ash venting processes directly with specific structures within the vent (Fig. 1f–h). In particular, here we describe veneered surfaces of in-situ fracture planes (Fig. 1g, h), which are the surface expression of ash venting pathways active during the prolonged hybrid phase of the eruption. By January 2014, the vent areas consisted of fractured obsidian lava with ubiquitous red fracture surfaces (Fig. 1g, h). These—as with tuffisite structures found in the distal flowfield[16]—correspond to sub-vent structures observed in 2012 (Fig. 1c–e)[9]. They range from smooth, curviplanar surfaces extending over several meters to complex smaller-scale surfaces that follow pre-existing lava carapace cooling joints or wend between brecciated lava clasts (Fig. 1g). In-situ fracture surfaces display prominent, predominantly vertical grooves and impact marks (Fig. 1g), but negligible evidence for shear displacement. Surfaces are coated by variably sintered, fine-grained ash veneers, ranging from dense, red indurated layers to dusty pink/yellow coatings that thicken into surface hollows and appear to be vertically extensive into the lava. Veneers are several microns to millimetres thick (Fig. 1g, h) and contain embedded, angular larger crystalline lava clasts ≤10 cm across, themselves coated by a veneer of glassy ash.

Four samples were analysed using scanning electron microscopy: AN1, AN2, CCTVAIP, and CCVP. Imagery of ash veneers reveals a diversity of particle morphologies, including vesicle-free, glassy ash shards and rounded fragments with grain diameters in the range

0.1–100 μm (Fig. 2a–c, e–g, i–k, m–o) and predominantly <1 μm (Fig. 2d, h, l, p). Ash morphologies range from angular (e.g. Fig. 2b, f, g, m), with conchoidal fracture surfaces, to near-spherical (e.g. Fig. 2e, j) and droplet-like with narrow necks between ash particles and other surfaces (Fig. 2c). In some cases, the ash particles are coalesced into smooth surface coatings (Fig. 2k, o). Clusters of strongly-sintered clasts ~10 μm across and aligned, millimetric pellet-like agglomerations that themselves consist of finer-grained ash. Pellets are aligned parallel to the ropy texture conspicuous on many surfaces (Fig. 2a). A thin outer layer of coarser clasts commonly adheres to hollows in the strongly-sintered ash veneer, whereas some veneer-free surfaces instead display scouring. We interpret these textures as representing varying degrees of sintering[28,29], which is most progressed in the finest-grained coating material where there is near-complete inter-particle porosity destruction[29] and where smooth surface coatings result (Fig. 2n, o). Larger particles, pellets, and aggregates are less well sintered with incipient necks representing only the early stages of sintering[29] (Fig. 2c, e). We infer the scouring to represent a combination of mechanical abrasion and corrosion of glass. As well as necking (Fig. 2c), other evidence points to the veneers being composed of a continuum of variably sintered particles adhered to a competent substrate: in Figs. 2b, 2e, for example, discrete particle shapes (platy and near-spherical, respectively) can be clearly observed, suggesting the earliest stages of contact sintering. In Fig. 2k, adjacent particles exhibit different stages of welding onto a larger grain. In Fig. 2o, the tumulus-like lumps (highlighted) represent the advanced stages of droplet sintering to a planar substrate.

Despite the variety of microtextures observed across the four samples (Fig. 2), including angular fragments such as highlighted in Fig. 2m, energy dispersive X-ray (EDX) analysis indicates that the veneers are broadly homogeneous in composition, revealed by elemental mapping (Fig. 3a–c) and point analyses (Fig. 3d–h). Angular fragments (Fig. 3d, f) are largely indistinguishable from more rounded and fluidal particles (Fig. 3g, h); the substrate (Fig. 3d, e) appears relatively depleted in mobile cations (Na, Al, K) relative to the variably sintered particles. The lack of additional elements such as S (which would be manifest as a peak in counts ~2.3 kEv) indicates that the particles are rhyolitic glass, with a minor plagioclase component consistent with phenocryst populations, but with no evidence of mineral precipitation. Additional EDX data are provided as Supplementary Information.

### Capture and sticking of volcanic droplets

The observation of variably sintered fine-ash veneers coating the margins of what were active ash-venting pathways can be described in the context of our current understanding of silicic explosive-effusive transitions[2,14]. Here, we distil three starting observations that can be used to underpin a description of the dynamics that form the observed ash-veneers, and help us to interpret their significance for eruptions. First, ash must have been generated down to the ultra-fine, sub-micron particle sizes observed in the sintered veneers (Fig. 2): in Fig. 4a, we plot the distribution of the particle radii, which range from $5.35 \times 10^{-8}$ to $2.96 \times 10^{-6}$ m, with a mean of $3.47 \times 10^{-7}$ m. Second, the particles must have been transported in a gas-ash dispersion (observed in ash-laden jets during the eruption and feeding the intermittent gas-ash plumes that characterise the hybrid explosive-effusive eruptive style; Fig. 1c–e). Third, the fine particles must have sintered rapidly to the fracture walls without rebounding or being re-entrained into the bypassing flow. From these observations, we posit that the grainsize of the material in the sintered surface deposits—extremely fine when compared with fall deposits derived from the same eruptive phases[30,31]—governed their preferential capture and sintering to the surface. This in turn requires an exceptional flow and depositional environment that would have fostered fines being preferentially transported to the walls and sequestered in the boundary layer flow at

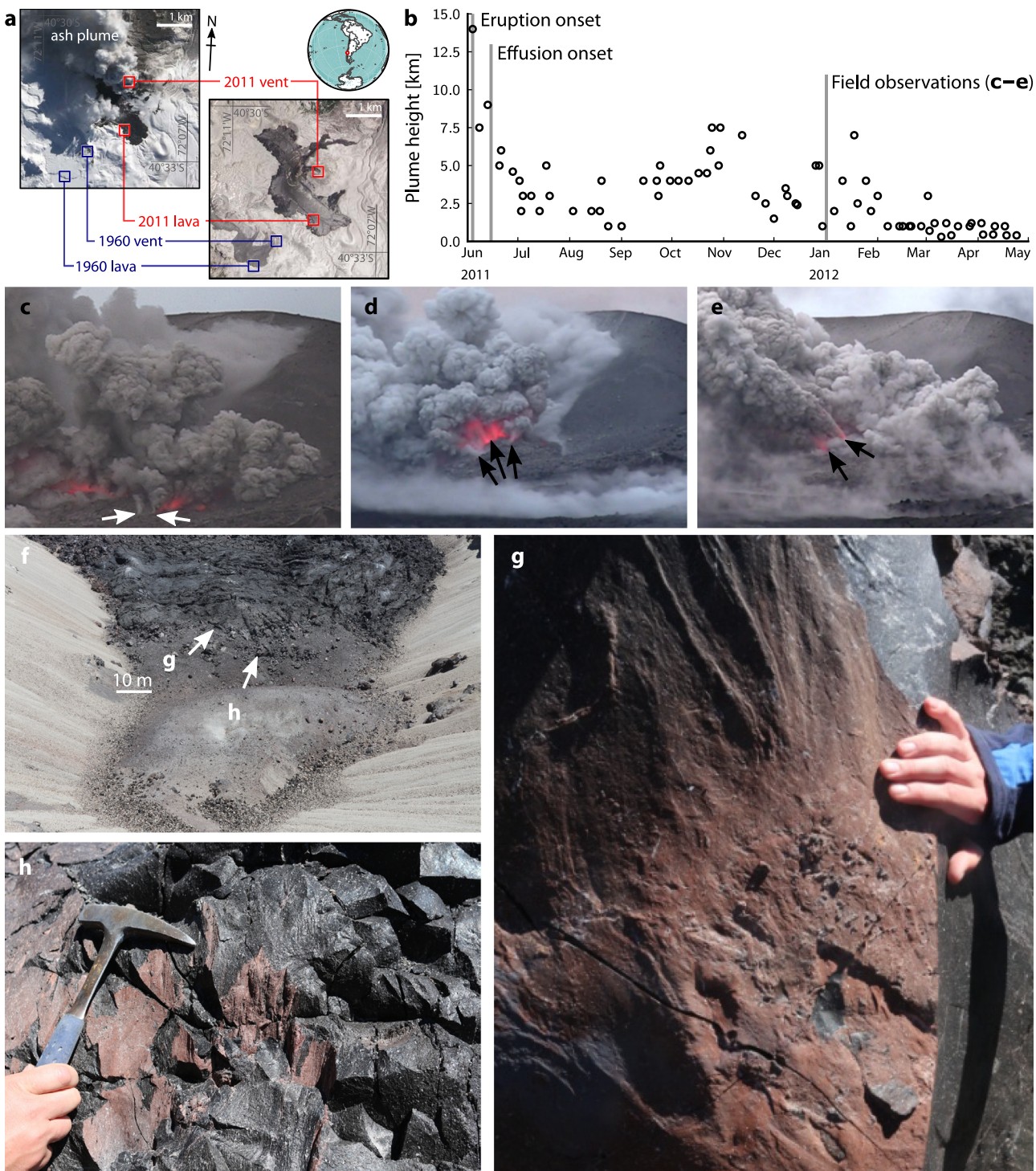

**Fig. 1 | Syn- and post-eruption observations at Cordón Caulle. a** Location of 1960 and 2011 eruptive vents; images from the Advanced Land Imager on NASA's Earth Observing-1 (EO-1) satellite (18/8/2011 [left]; 13/1/2013 [right]). Satellite data courtesy of NASA's Earth Observatory https://earthobservatory.nasa.gov/. **b** Plume height above sea level over time, from onset in July 2011 until May 2012 (see ref. 27). Onset of effusion (i.e. hybrid phase) is highlighted, as is the date of on-site observations shown in **c–e. c–e** Video frames recorded in January 2012 showing the main Cordón Caulle vent region. Arrows highlight discrete, semi-transient ash vents. **f** Vent region in 2014. **g** Detail of reddish veneer coating curviplanar ash-vent surface (see (**f**) for location). **h** As (**g**), detailing ash veneer on hackly lava surface.

the wall, despite there being proximal and simultaneous high-energy flows capable of transporting larger particles beyond the walls and into the erupting plume. We use these conceptual starting points to underpin a scaling analysis of the transport and deposition regimes involved in order to gain insight in to the timescales and mechanisms involved.

The first question is whether or not particles in turbulent flow (see Methods) will contact the wall, or be transported through the fracture without interacting with the wall. The characteristic timescale of motion of a particle of diameter $d$ in a flow is $\lambda_p = \rho d^2/(18\mu_f)$, where $\rho$ is the particle density and $\mu_f$ the dynamic viscosity of the carrier phase. The response of the particle to flow turbulence is described by the

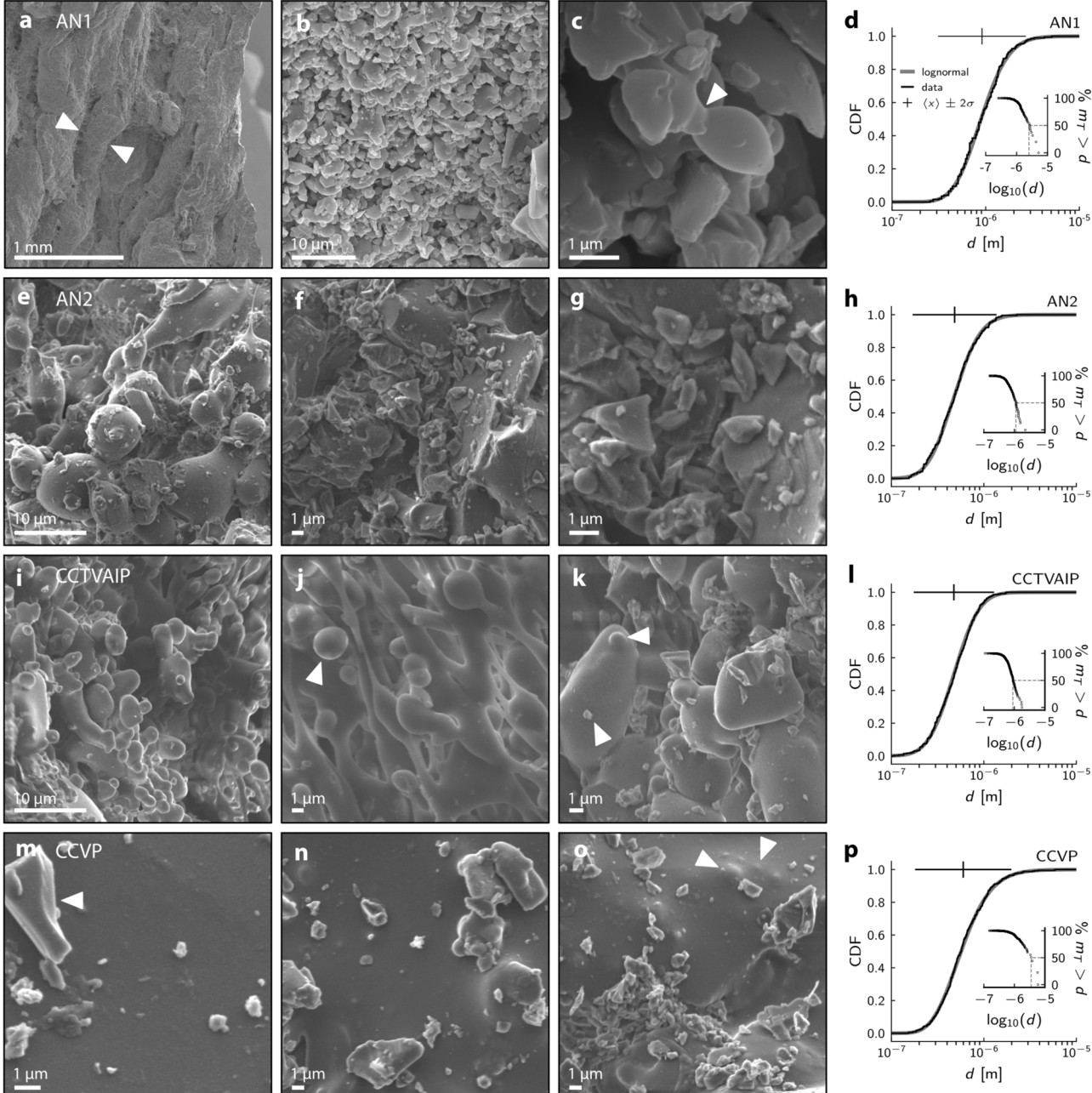

**Fig. 2 | Particle sizes in sintered veneers on in-vent lava fractures that fed ash-venting during hybrid explosive-effusive eruptions. a–c** SEM images of fracture-wall surface textures in sample AN1. Ropy textures are highlighted in **a**; evidence of neck formation highlighted in **c**. **d** Cumulative distribution (cdf) of particle diameters $d$, compared with a lognormal function $\left[1 + \mathrm{erf}\left(\ln x - \langle x \rangle / \sigma \sqrt{2}\right)\right]/2$ defined by the mean $\langle x \rangle$ and standard deviation $\sigma$ of the logarithm of the data $x$. Density distribution of particles shown as $\langle x \rangle$ (vertical tick) and $2\sigma$ range (horizontal line). Inset shows the mass proportion (assuming spherical particles and homogeneous melt density) of particles greater than a given diameter. **e**, **g** Microtextures observed in sample AN2. **h** As (**d**), for sample AN2. **i–k** Microtextures in sample CCTVAIP. Near-spherical particle is highlighted in (**j**). In (**k**), arrows indicate both angular fragment and well-sintered particle adhered to larger clast. **l** As **d**, for sample CCTVAIP. **m–o** Microtextures in sample CCVP. Discrete angular fragment highlighted in **m**. Remnants of sintering highlighted in **o**. **p** As (**d**), for sample CCVP.

Stokes number St, which is the ratio of $\lambda_p$ to a characteristic fluid timescale $\lambda_f$, in turn a function of a given eddy size and velocity within the flow. Therefore, $\mathrm{St} = \rho d^2/(18\mu_f \lambda_f)$ where $\mathrm{St} \gg 1$ represents particles that are not well-coupled to the flow and may impact the walls. The Stokes number also correlates with the organisation of particles within an eddy, controlling their local concentration or dispersal as a function of their size[32].

We can place bounds on the maximum and minimum eddy sizes possible in turbulent flow as corresponding to either the fracture aperture $L$ (upper bound) or the Kolmogorov lengthscale $\delta_K$ (lower bound), respectively. First, the upper bound on the maximum eddy size is associated with a maximum fluid timescale $\lambda_f = \lambda_O = L/\langle u \rangle$, where $\langle u \rangle$ is the mean velocity of the flow. Second, the minimum eddy size is associated with the Kolmogorov timescale $\lambda_f = \lambda_K = \sqrt{\nu/\varepsilon}$, where $\nu$ is the kinematic viscosity of the fluid, $\varepsilon = \nu^3/\delta_K^4$ is the average rate of dissipation of turbulence kinetic energy per unit mass, and $\delta_K = \nu/\langle u \rangle$. At high Reynolds number (Methods), turbulence generates eddies from the scale of $\delta_K$ to $L$, meaning that there exists a range of potential St associated with any given particle size, depending on the eddy size local to the particle. In $u$-$R$ space (Fig. 4b), we can delineate three

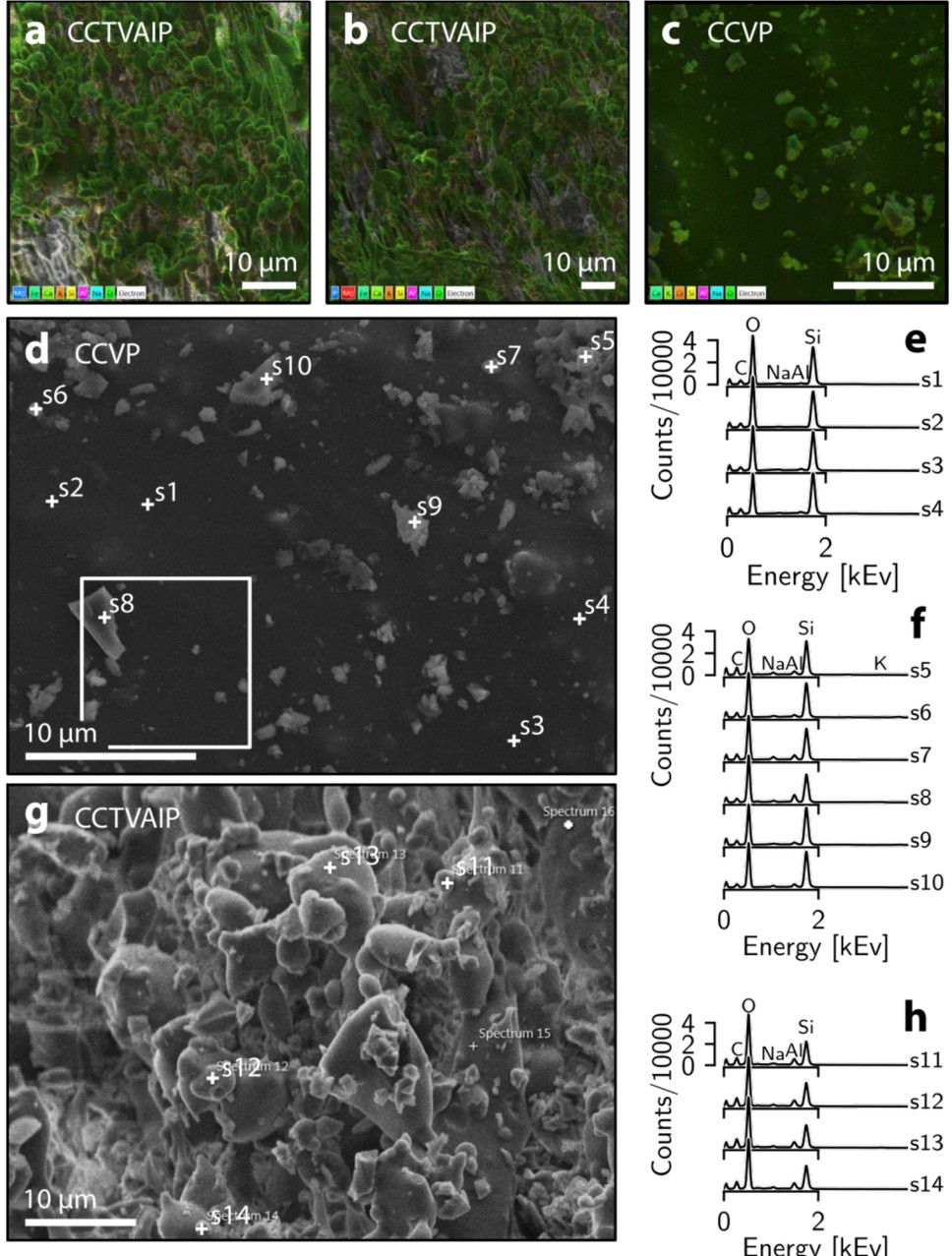

**Fig. 3 | Geochemistry of the sintered veneers. a, b** Energy-dispersive X-ray spectroscopy (EDX) element maps of sample CCTVAIP, with Si, O, and additional elements overlain on an SEM image. **c** As (**a**), for sample CCVP. **d** Point EDX analyses for sample CCVP, with spectra for points s1–s4 (the substrate) shown in **e**, and points s5–s10 (discrete particles) in **f**. Peaks for C, O, Na, Al, Si, and K are highlighted. Note white box showing area of Fig. 2m. **g** As (**d**), for sample CCTVAIP. Spectra for points s11–s14 are shown in **h**.

regimes of particle response to turbulent flow: St ≫ 1, wherein particles cross streamlines and may impact walls; St ≪ 1, wherein particles follow streamlines; and St ~1, wherein particles will tend to concentrate (cluster) at the periphery of eddies, potentially decoupling from the gyratory flowpath[33] (Fig. 4b). In detail, the condition St = 1 may be met depending on the eddy size (between $\delta_K$ and $L$). In Fig. 4b we plot these regimes for the fracture width $L = 0.01$ m measured in the vent at Cordón Caulle. The unshaded region indicates the range of $\langle u \rangle$ and $R$ where it is possible for St to equal 1 (dependent on eddy size): outside this region, St must be lower than unity (below the St = $\lambda_p/\lambda_K$ line) or greater than unity (above the St = $\lambda_p/\lambda_O$ line). Notably, the St ~1 regime corresponds well with the observed particle size distribution at velocities of the order of $\langle u \rangle$ (Fig. 4a). Experiments and simulations illustrate that when St ~1, non-uniform particle distribution is observed[32,33]. The fact that the observed grainsize distribution falls within the range whereby St ~1 suggests that this size fraction in particular is dynamically decoupled from the carrier phase and will cluster in the flow. Particles that are not large enough to be on truly ballistic trajectories (St ≫ 1) and not small enough to follow the smallest eddy flows in an effectively passive manner (St ≪ 1) exhibit a particular tendency to migrate from regions of high turbulence to low-turbulence-intensity regions[34,35] via a phenomenon termed turbophoresis. Turbophoresis serves to segregate particles axially toward the wall region, and is facilitated by a viscous boundary layer at the wall that in turn arises from the velocity differential between $\langle u \rangle$ and $u_T$ (the boundary friction velocity[36]; see Methods). We observe a diversity of

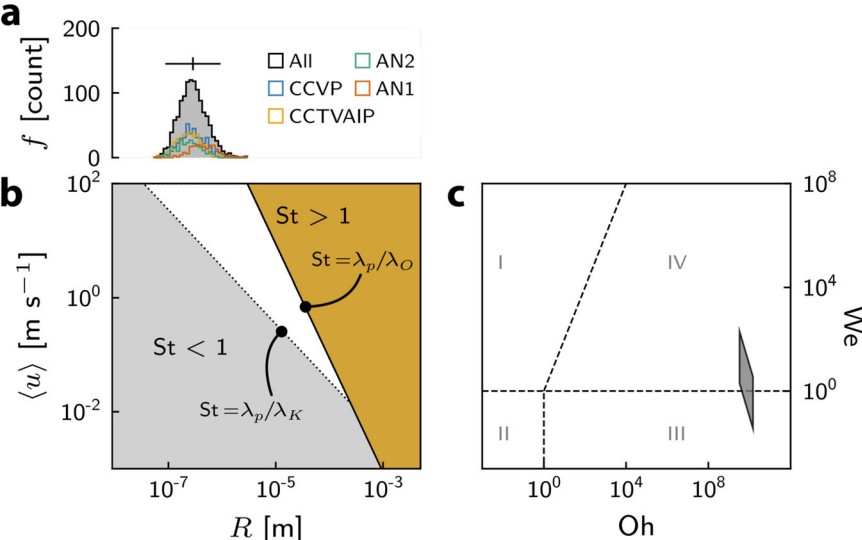

**Fig. 4 | Droplet dynamic regimes. a** Histograms of particle radius $R$ (shown as frequency counts $f$) for 4 samples shown in Fig. 2, and the amalgamated data ("All"). Mean and $2\sigma$ range is shown for combined data. **b** Stokes number regimes for a 0.01 m wall-bounded flow as a function of velocity and particle radius. The different regimes are delineated according to the maximum and minimum eddy sizes in the flow ($\lambda_O$ and $\lambda_K$, respectively). **c** Weber–Ohnesorge plane for the range of particle sizes, droplet density, surface tension, viscosity, and velocity (see text). The four regimes are: I (inertial impact–driven); II (inertial capillary–driven); III (viscous capillary–driven); IV (viscous impact–driven)[43]. At $\langle u \rangle$ ~ 100 m s⁻¹, We≫1; at $\langle u \rangle$ ~ 10 m s⁻¹, We ~ 1.

particle shapes in our samples (Fig. 2); however, we do not account for this explicitly, as recent research suggests that while particle shape has a large effect on particle velocity and rotational dynamics, it has comparatively little effect on translational dynamics (i.e. movement from the centreline to the wall)[37].

Having determined that a portion of the particles are at St > 1 and can undergo a particle-wall interaction, we now determine the dynamics of that interaction by considering the particles—at magmatic temperatures—to effectively be viscous droplets. At the droplet scale, isothermal dynamics can be characterised by the Eötvös, Ohnesorge, and Weber numbers (Eo, Oh, and We, respectively, see Methods). Taking a representative value of surface tension $\Gamma = 0.36$ N m⁻¹ for degassed volcanic particles[38–40], characteristic particle radii $R$ of order $5.35 \times 10^{-8} \leq R \leq 2.96 \times 10^{-6}$ m (Fig. 4a) yield Eo ≪ 1, indicating that the effect of body forces arising from gravity are negligible. In addition, Oh ≫ $\sqrt{\text{We}}$ for temperature-dependent droplet viscosity $\mu(T)$ of $1.56 \times 10^{8}$ Pa s [refs. 41,42], where $T = 900$ °C is assumed to be a representative magmatic temperature based on geothermometry[41]. Taking $\langle u \rangle = 100$ m s⁻¹ to be the characteristic exit velocity of particles based on image velocimetry[9] and $u_\theta$ to be the impact angle–scaled velocity (Methods), We is typically >1. In Fig. 4c we plot upper and lower bounds on the Oh-We plane[43]. The majority of feasible values plot in regime IV, in which the force driving the impact droplet-wall interaction is the impact force itself, capillary forces at the contact line of the droplet are negligible, and the force resisting the impact interaction arises from the viscosity of the droplet. Consequently, we can establish the characteristic timescale of the droplet-wall interaction as $\lambda_d = \mu(\rho u_\theta^2)^{-1}$, where $\rho$ is droplet density[44].

Finally, we address whether or not a particle can sinter on the timescale available in the interaction. To assess this, we appeal to the observation that in all droplet-wall interactions reproduced experimentally, interaction will result in sticking if the droplet is viscous at the time of impact[43,45]. As more droplets interact and stick, the sintering process that creates a less porous and stronger deposit layer (i.e. the "porosity destruction" inferred above) is governed by the timescale for particle viscous sintering—driven by surface tension at the interface of the particles[46]—and is $\lambda_\Gamma = R\mu/\Gamma$, yielding timescales ~55–1104 s. These values are similar to the interaction times $7 \leq \lambda_d \leq 647$ s for

$100 \geq u_\theta \geq 10$ m s⁻¹, and are also generally within the timescale of ash-jetting episodes (seconds to tens of seconds[9]). We note that any nonzero pressure differential acting on the droplets in excess of the gas pressure will serve to accelerate the sintering process[2,28,29]. Recent work has shown that small rhyolitic ash particles ($R$ ~ 10⁻⁶ m and smaller) will sinter more readily than larger particles, despite a smaller size fraction being more thoroughly degassed at the Earth's surface and therefore having a higher viscosity[2].

It is compelling that the sintering timescale is generally less than the droplet-wall interaction timescale (i.e. $\lambda_\Gamma \leq \lambda_d$); however, we highlight that this is not a strictly necessary condition for droplets to stick to the wall in this system: molten silicic droplets will, in all likelihood, stick when interacting with a hot surface[43,45]. This means that $\lambda_d$ is therefore indicative of the initial stick and spread dynamics. As more droplets stick and accumulate a surface deposit, the sintering time $\lambda_\Gamma$ becomes more relevant, revealing the most conservative time required for the deposit to densify to a non-porous state. In Fig. 2, it is clear that particles have variably undergone full sintering (cf. Fig. 2b, j), and that where individual particles can be seen, they have only variably spread onto the substrate they adhere to. This is consistent with our finding that $\lambda_\Gamma$ and $\lambda_d$ are predicted to be of a similar order of magnitude, and implies that different regions of these surfaces are likely to be at different temperatures, accounting for the observed variability in texture. Both $\lambda_\Gamma$ and $\lambda_d$ have a linear dependence on the droplet viscosity, such that any syn-eruptive cooling during transport that would serve to increase these timescales, would do so proportionally to both.

Particle organisation and entrapment could be enhanced by electrostatic forces[47,48], particle-dispersion heating by any particle-particle and particle-wall friction at high particle volume fractions, the growth of viscous boundary layers during waning flow, and large wall asperities that host low velocity eddies. Taken together, this analysis demonstrates (1) the turbulent motion of the carrier phase should migrate fines towards the fracture/conduit walls, in particular (2) those particles within the $10^{-8} \leq R \leq 10^{-5}$ m range are most likely to decouple from the carrier phase and subsequently impact the walls and interact, and (3) that interacting particles can stick and sinter. Critically, our analysis highlights that the small grainsize is the key parameter at each step of preferential segregation, transport, and sticking (Fig. 5b).

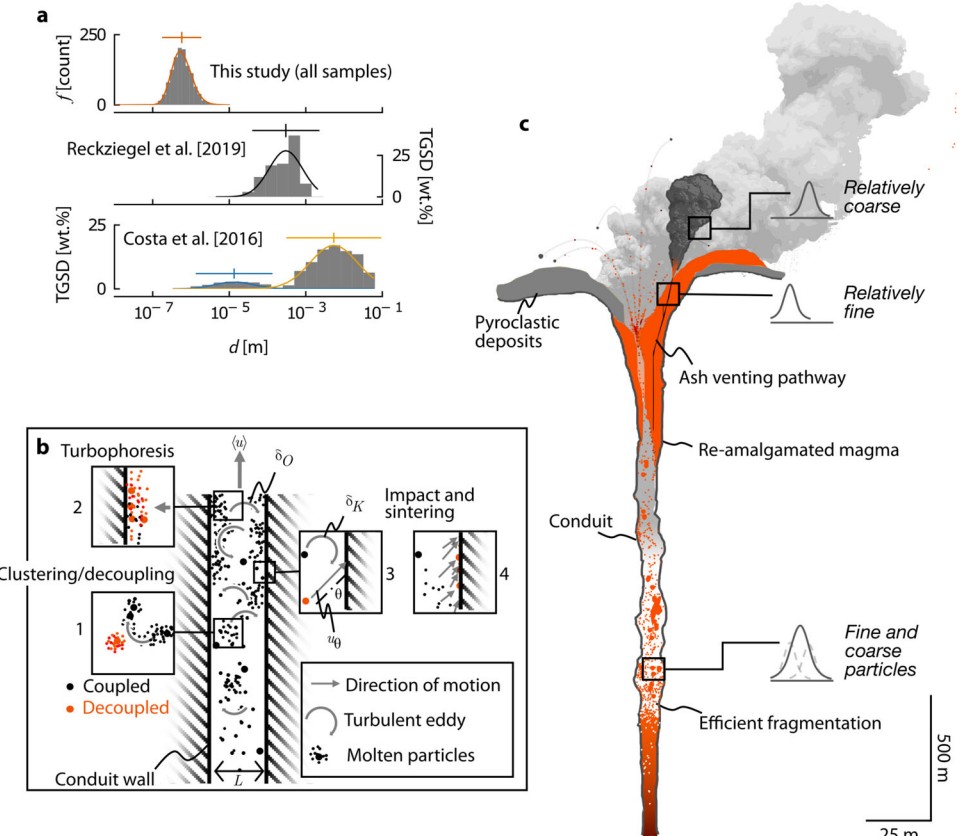

**Fig. 5 | A grain size fractionation model for Cordón Caulle volcano. a** Compiled data in this study reflect a captured in-conduit fine ash fraction, characterised by a mean (as-measured) diameter $d$ of $9.94 \times 10^{-7}$ m. For each dataset, best-fit log-normal curves have been overlain. Distribution assumed by Reckziegel et al.[31] are approximately unimodal and described by mean of $d = 3.10 \times 10^{-4}$ m (based on lognormal assumption of the ash mass fraction). Data of Costa et al.[30], reconstructed from field data, are bimodal, described by lognormal peaks at $d = 1.34 \times 10^{-5}$ and $d = 5.39 \times 10^{-3}$ m. Mean and $\pm 2\sigma$ range are highlighted for data of this study and ref. 31; these values are shown for each of the peaks of ref. 30. Note different bin size and y axes between panels. **b** In-conduit processes summarised schematically. Magmatic particles are transported in a gas phase at a mean velocity of $\langle u \rangle$. Flow turbulence results in clustering and decoupling of particles (1) of St $\geq 1$ from the eddy motion, in addition to turbophoresis (2) which results in elevated concentration of particles at the wall. Particles may then impact the wall (3) at velocity $u_\theta$—dependent on $\theta$—then sinter rapidly (4). **c** Energetic fragmentation within the conduit generates a population of both fine and coarse particles; due to the processes outlined in **b**, the relatively fine fraction is preferentially captured at the walls of ash vents; the emitted ash therefore predominantly reflects the coarser fraction of the original population. Conceptual model of the shallow subsurface depicted in **c** is from the so-called cryptic fragmentation model for silicic hybrid and effusive eruptions described in ref. 2. Scales are approximate, as the crater— and presumably the conduit—diameter and morphology changed throughout the course of the eruption.

## Grain size fractionation

In this work, we show that fine volcanic ash particles have stuck to fracture planes during ash venting, and were therefore captured and sintered while capable of viscous deformation. It is small particles at the low end of the high-Stokes-number regime that will be preferentially transported by turbophoresis and therefore sintered to the walls; as such, fine particles are fractionated from the TGSD prior to eruption into the plume. TGSD is used as a window into pre- and syn-eruptive fragmentation[49], and to directly infer sub-surface magmatic processes[30]. Conventionally, reconstructed TGSDs inform—albeit indirectly—the mechanisms, efficiency, and depth of magma fragmentation and the total explosive energy release[50–52]. Moreover, TGSDs are used to determine initial tephra size distributions, a critical source term for ash dispersal models[30,53]. Fundamentally, such estimates rely on the basic assumption that reconstructed TGSDs are directly related to these source parameters; however, secondary processes may modify the size distribution of primary fragmentation products[13]. The TGSD from the Córdon Caulle 2011 eruption has been reconstructed or otherwise estimated by various methods[17,30,31]. These estimates differ, both in terms of range and modality (Fig. 5a); importantly however, there is little overlap with the size distribution associated with the features of this study. The fine ash fraction observed here therefore reflects more energetic fragmentation[50] than the relatively larger fraction captured by post-emplacement approximations (Fig. 5c), which cannot fully capture the energetics or mechanics of fragmentation at depth both due to fine-ash entrapment within the conduit, and to syn- and post-eruption processes including aggregation, winnowing, and remobilisation[54]. The exponential nature of hydrodynamic fragmentation models means that extrapolating to submicron scales yields implausibly high fragmentation energies. Primary generation of fine and ultra-fine ash must therefore be the result of a brittle fragmentation mechanism, for which there is no generally accepted theory[50]. It is our hope that future generations of fragmentation models account explicitly for the potential generation of submicron-scale particles.

The capture and sintering of particles in the volcanic sub-surface has recently been suggested as a mechanism to assemble silicic lava bodies[2,14] as a whole. A tantalising implication of this model is that the process of within-conduit sticking and sintering is volumetrically significant relative to the whole erupted volume, and may mean that the fines production at depth (1) is volumetrically substantial, and (2) represents a significant proportion of the explosive energy[50]. The downstream implication is that the fragmentation energy may be higher than can be predicted by observations of grain sizes in fall

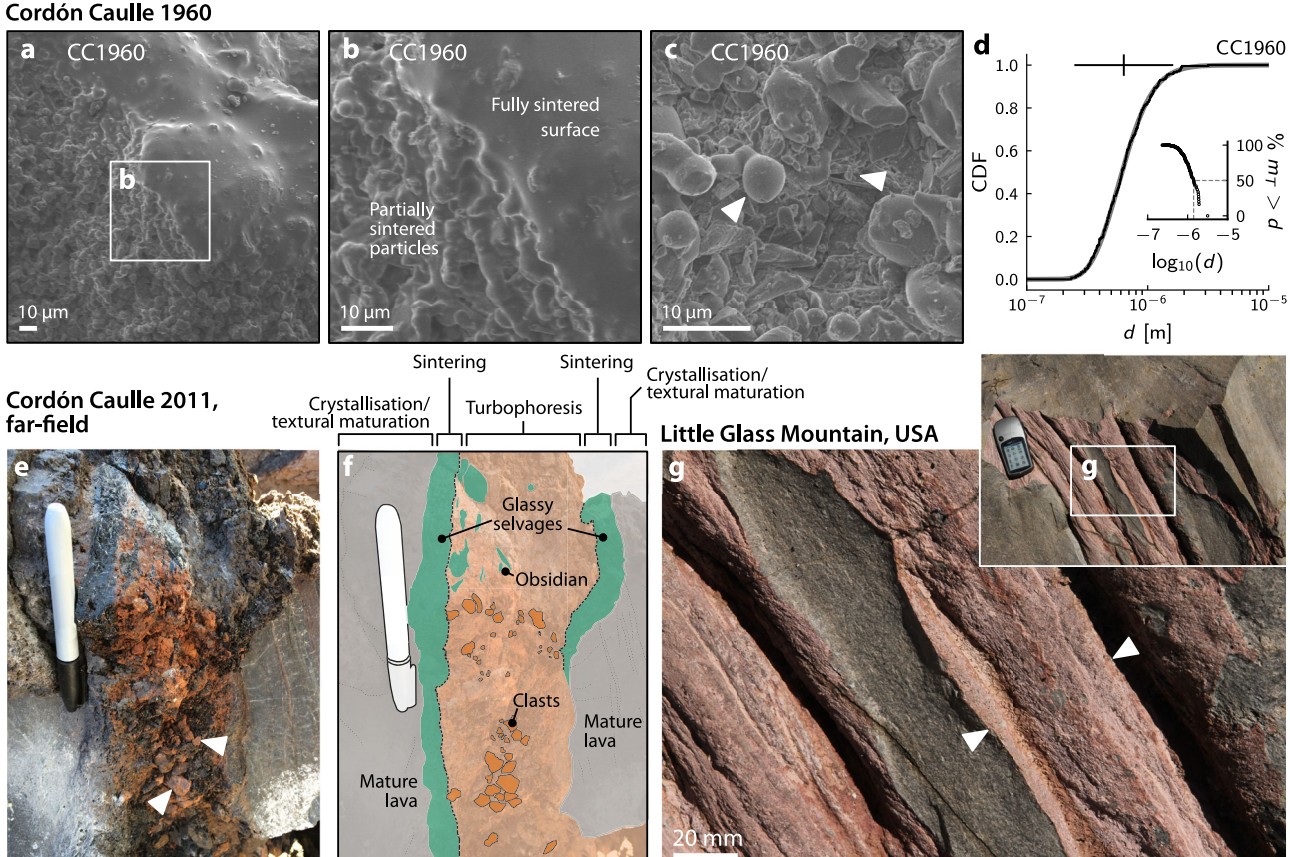

**Fig. 6 | Extended field observations. a–c** SEM images of sample CC1960, collected from Cordón Caulle 1960 CE eruptive vent. In **c**, both near-spherical and angular particles can be observed. **d** Particle size distribution for sample CC1960, as in Fig. 2. **e** Detail of veneered fracture surface distal from the main vent at Cordón Caulle. Arrows highlight millimetric-to-centimetric clasts sintered to the fracture surface. Marker pen is ~14 cm. **f** Schematic interpretation of **e**, indicating laterally juxtaposed zones that had experienced various primary operative processes during

emplacement (turbophoresis, sintering, crystallisation/textural maturation). Note that a planar fracture conduit geometry facilitates this ordering of textural zones, resulting in a local time-transgressive sequence of lava production. **g** Detail of a veneered fracture surface at Little Glass Mountain (Medicine Lake volcano, northern California). Reddish ash-coated surface is highlighted by the arrows. Inset shows ash-coated surface in context of exposed fracture plane.

deposits—reinforced by recognition that many silicic pumice clasts are in fact sintered agglomerates[13,55]. Other processes—including shear or abrasion mechanisms[56,57] and related in-conduit comminution processes[58]—act to divorce grain size distributions from the magmatic source conditions, increasing the fines proportion relative to the size distribution produced at true magmatic fragmentation at depth. By contrast, the process we invoke to explain our observations acts in the opposite direction, sequestering fine grains and leaving a relatively coarser grain size distribution to erupt in the plume than that produced at magmatic fragmentation (Fig. 5c). Note that—unlike TGSD modification driven by post-emplacement agglutination (proposed for Hawaiian-style basaltic systems[59])—we anticipate in-conduit turbophoresis and sintering to be one of the earliest operative syneruptive processes due to the colocation of the fragmentation and particle migration mechanisms. In concert, these effects may strongly decouple the erupted pyroclast and ash size distribution from the magmatic source conditions.

### Implications for silicic eruption dynamics

The Cordón Caulle eruption allowed the observation of contemporaneous explosive and effusive behaviour, a phenomenon which has since been interpreted as a product of continued in-conduit explosive fragmentation[16] and variable reaggregation of clasts via sintering[2]. Our data further support this hypothesis. The measured size distribution is indicative of highly efficient fragmentation[51,60] and means that these ashy veneers record

fragmentation processes at Cordón Caulle that are not captured through existing TGSD reconstructions (Fig. 5a). It is likely that energetic magmatic fragmentation was sustained throughout the hybrid phases of the Cordón Caulle eruption[2], with in-conduit organisation and sintering effectively removing the evidence of this process from the erupted componentry. This is supported by evidence of ash-coated fracture planes distal to the main vent (Fig. 6e). In order to fully capture the dynamics of hybrid silicic volcanism—critical to hazard mitigation—models of eruptive activity must couple the influence of continuous, energetic fragmentation and the preferential in-conduit entrapment of ultrafine ash.

There is mounting evidence that silicic eruptions are characterised by explosive fragmentation and variable in-conduit re-aggregation of magma[2,3,11,13]; the fact that the vent "nozzles" examined here preserve evidence of essentially the same processes—including the formation of variably sintered particle clusters (e.g. Fig. 2o)—suggests that these mechanisms may be self-similar across spatial scales. We show that particle capture and sticking are functions of conduit or fracture width, which may itself evolve as the walls are progressively coated with a sequestered ash fraction. A key implication of this model is that the transport of ash from its fragmentation source to the vent not only modifies the subsequent recorded componentry, but may in fact directly influence eruptive dynamics by altering variables such as conduit width, particle density, and bulk viscosity. In addition, our model predicts that gradual waning of vent flux (i.e. a decrease in $\langle u \rangle$: Fig. 4b) may facilitate the capture of a progressively larger size

fraction—as observed elsewhere in the flowfield (Fig. 6e)—potentially to the point of occluding the fracture entirely. Figure 6e (interpreted in Fig. 6f) illustrates that the walls of actively venting fractures are the sites of particle capture, sintering, and ultimate densification of ash into coherent lava. This process may be efficient and operate over quick enough timescales to create zones of accretionary lava at the margins of vents (Fig. 6e, f). We have focussed here on ash-coated nozzle structures at the main 2011 vent of Cordón Caulle; we note, however, that not only do comparable features exist throughout the flowfield (Fig. 6e), comparable microtextures (Fig. 6a–c) and particle size distributions (Fig. 6d) are revealed by samples from the 1960 eruptive vent (Fig. 1a). Notably, sample CC1960 exhibits clear sintering textures, including a transition from densely packed partially welded samples (both spherical and angular: Fig. 6c) to a fully cohesive sintered surface (Fig. 6a, b). Finally, we highlight that these textures exist at other rhyolitic volcanoes: Fig. 6g shows evidence of similarly veneered fracture surfaces at Medicine Lake volcano (USA). This fact suggests that the processes described are ubiquitous throughout rhyolitic systems and occur throughout hybrid activity—even in the earliest stages. Considering post-fragmentation processes in terms of a hot, particle-laden (i.e. dusty) gas flow thus represents a frontier in silicic eruption modelling.

## Methods
### Plume height
Measurements of plume height were collected by El Servicio Nacional de Geología y Minería (SERNAGEOMIN) and Observatorio Volcanológico de los Andes del Sur (OVDAS)[27].

### Videography
Activity was recorded in January 2012 using a Canon XF105 video camera at 25 FPS in visible light mode. For further details, the reader is referred to ref. 9.

### Microscopy and grainsize analysis
Scanning Electron Microscope (SEM) images were collected at the University of Liverpool, with a Philips XL30 tungsten filament SEM used on carbon-coated standard petrographic thin sections, and at Lancaster University Chemistry department with a JEOL JSM-7800F SEM. The grain sizes discussed here are generally too small for many conventional analyses. In order to determine a grain size distribution, the semimajor grain axis from representative 2D SEM images were manually measured with ImageJ. These diameters were converted to a mass distribution by assuming a spherical particle geometry, with melt density as defined previously. We note that volume (and thus, mass) estimates are approximate, due to the variability in particle morphology, which cannot be accurately determined in two dimensions. Mass distributions are used illustratively, and the measured grain diameters are instead used in any further analyses. As we use the semimajor axis, particle diameters are nominally maxima, notwithstanding their extension in 3D. All particle size data shown in Fig. 2d, h, l, p were obtained at 4000–4300 magnification, and all data in Fig. 6d at 3700–4300 magnification: annotated SEM images are provided as Supplementary Fig. 1. The number and size of particles that can be classified in this way is dependent on image size and resolution; accordingly, in order to ensure the total particle population was appropriately and representatively sampled, we also measured particles using a wider range of magnifications. Decreasing and increasing the magnification to 500 × and 10,000 ×, respectively, did not appreciably affect the fraction of largest or smallest particles observed. We obtain log-normal particle size distributions for each sample, indicating that we are capturing a full distribution. Altogether, we show data from 1467 measured particles.

### Geochemical analysis
Energy dispersive X-ray spectroscopy (EDX) was used to create element maps of thin-sections of samples CCTVAIP and CCVP. In addition, 14 point analyses were also performed on the samples, in order to target the substrate material and particles of varying morphology. EDX maps are provided as Supplementary Figs. 2–5.

### Dimensionless regime calculation
Flow turbulence in a wall-bounded particle-laden gas flow is characterised by the Reynolds number Re $= \langle u \rangle L / \nu$, where $\langle u \rangle$ is the mean velocity of the flow, $L$ a characteristic lengthscale (in this case the fracture width), and $\nu$ the kinematic viscosity of the fluid. In flow through a slot geometry, fully developed turbulence is expected when Re $\geq 2900$ [ref. 36]. Given estimates of $\langle u \rangle$ -100 m s$^{-1}$ [ref. 9], $0.01 < L < 1$ m [e.g. refs. 61, 62], and $\nu(T) = 1.4 \times 10^{-4}$ m$^2$ s$^{-1}$, gas venting through fractures during the Cordón-Caulle eruption can be assumed to be highly turbulent ($7.14 \times 10^3 \lesssim$ Re $\lesssim 7.14 \times 10^5$).

At the droplet scale, isothermal dynamics can be characterised by the Eötvös (Eo), Ohnesorge (Oh), and Weber (We) numbers. Eo defines the ratio of gravitational and surface tension forces: Eo $= \rho g R^2 / \Gamma$, where $\rho$ is droplet density, $g$ is gravitational acceleration, $R$ is the particle radius ($d/2$), and $\Gamma$ is interfacial tension at the wall. Oh characterises the relative importance of viscous and inertial forces arising from capillary flow: Oh $= \mu / \sqrt{\rho \Gamma R}$, where $\mu$ is droplet viscosity. Weber number We scales the effects of inertia and surface tension forces: We $= \rho R u_\theta^2 / \Gamma$, where $u_\theta$ is the relative droplet velocity, here scaled by the impact angle $\theta$ such that $u_\theta = \cos\theta \langle u \rangle$. Note that for a uniform distribution of potential impact angles $0 < \theta < \pi$, 50% of $u_\theta$ will be 87 m s$^{-1}$ or greater: only at very high 'glancing' impact angles does $u_\theta$ approach zero. We $\gg 1$ and We $\ll 1$ denote the inertial and capillary fields respectively; for Oh, the critical value is given by $\sqrt{We}$ such that Oh $\gg \sqrt{We}$ and Oh $\ll \sqrt{We}$ denote the viscous and inertial fields, respectively[43]. We note that droplet splashing can occur in regime I, and is typical of low viscosity droplets such as produced at basaltic eruption conditions (cf. ref. 63), and/or very high impact velocities.

### Boundary layer thickness calculation
In the region of fluid adjacent to the wall, there is a sub-layer in which the velocity of the bypassing fluid drops to lower values compared with the centre-line velocity. In order to assess the extent to which this sub-layer is laminar and represents a viscous boundary layer of appreciable thickness, we scale the dimensionless normal distance from the wall as $\bar{y} = y u_T / \nu$, where $y$ is the dimensional wall-normal direction, and $u_T$ is the boundary friction velocity[36]. To a first-order approximation, the viscous sub-layer occurs at $\bar{y} < 10$, a region in which the velocity profile is approximately linear $u(y) = \bar{y} u_T$ (ref. 36). This scaling implicates $u_T$ as the key unknown, and while the standard scaling is $u_T = \sqrt{\tau_w / \rho_f}$, where $\rho_f$ is the fluid density, this still leaves $\tau_w$ as an unknown[36]. Instead, we use the approximation $u_T \approx 0.05 \langle u \rangle$ for fully developed turbulence, which implies that the thickness of the region at $\bar{y} < 10$ is at $y_v = 200 \nu / \langle u \rangle$. Given the bounds on $\langle u \rangle$ and $\nu$ used earlier, this results in a minimum viscous boundary layer of thickness $y_v \approx 2.8 \times 10^{-4}$ m. This is larger than both the surface roughness lengthscale, and the particle sizes observed at the surface (Fig. 2), implying that the flow in this region will be hydraulically smooth. We note that this simple scaling implies that $y_v \propto \langle u \rangle^{-1}$, for which waning flow velocities would result in thicker boundary layers and more opportunity for particle entrainment and capture.

## Data availability
All data generated in this study are provided at the following repository: https://github.com/jifarquharson/cordon-caulle-ash-vents/, archived via Zenodo: Jamie Farquharson. (2022). jifarquharson/cordon-caulle-ash-vents: (v1.0). *Zenodo.* https://doi.org/10.5281/zenodo.

6815907. Additional SEM images and EDX element maps are provided as Supplementary Information.

## Code availability

Python code used to analyse and plot data are provided in a Jupyter Notebook at the following repository: https://github.com/jifarquharson/cordon-caulle-ash-vents/, archived via Zenodo: Jamie Farquharson. (2022). jifarquharson/cordon-caulle-ash-vents: (v1.0). *Zenodo*. https://doi.org/10.5281/zenodo.6815907.

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

## Acknowledgements

H.T. and J.F. were supported by a Royal Society University Research Fellowship UF140716 and fieldwork by F.W. was financially supported by the EC-FP7 Vuelco project. H.T., J.C., and C.I.S. thank Alex Barria and Miguel Obando for logistical assistance in the field. H.U. was supported through an Envision PhD studentship. Carmel Pinnington and Sara Baldock assisted with SEM imaging. We thank Lionel Wilson and Duncan Woodcock for discussion of particle transfer through pathways.

## Author contributions

H.T. conceived the study, obtained SEM imagery and EDX data, and—alongside F.B.W., J.C., and C.I.S.—collected observations and samples in the field. J.I.F. wrote the manuscript, created the figures, and analysed the data along with F.B.W. H.U. collected the grainsize data. All authors contributed to the writing of the manuscript.

## Competing interests

The authors declare no competing interests.
