## [Peer Review File · Nature Communications]

In-conduit capture of sub-micron volcanic ash particles via turbophoresis and sinteringEditorial Note: Parts of this Peer Review File have been redacted as indicated to maintain the confidentiality of unpublished data, to remove third-party material where no permission to publish could be obtained, and to protect the confidentiality of suggested referees.

REVIEWER COMMENTS

Reviewer #1 (Remarks to the Author):

Reviewer Report

Key results

Farquharson and authors propose a model in which submicron-scale ash particles are preferentially removed from the main flow via turbophoresis and subsequently agglomerate on conduit margins. Their model is based on physical field observations of the 2011-2013 eruption of Cordon Caulle and its resulting products, the latter which notably includes ash-veneered obsidian lava – the crux of their observational argument. Like tuffisites, they interpret such veneers represent upper conduit ash venting pathways. Feasibility of such interpretation of ash migration is supported by inner conduit process timescales and scaling arguments.

Validity

The authors propose a model that furthers strengthens arguments for upper conduit separation/amalgamation of pyroclastic products. Their study neatly fits in with others' work on ash and lapilli aggregation. I see no major flaws which should prohibit its publication, but I do have some suggestions worth considering mainly as noted in the commented manuscript.

Significance

This manuscript is of particular importance to the physical volcanology community, but has broader implications. It underscores the necessity of revisiting whether total grain size distribution (TGSD), as well as the erupted componentry, accurately capture inner-conduit processes. Notably, such ash depletion would remove fine material from the fallout TGSD, and as such result in inaccurate inputs for hazard models. Coupled with other product modification (e.g. secondary fragmentation) TGSDs increasingly seem to reflect less and less the state of magma at fragmentation.

An additional implication of their argument – which fits in the Wadsworth-pyroclastic-origin-of-effusive-products arguments – is that we are severely understanding the amount of fine material produced in the conduit, as the subsequent eruption of obsidian is volumetrically significant in context with the fallout phase. These ash veneers represent yet another fine pyroclastic component found in the effusive phase.

Data and Methodology

While I see no glaring issues with the methodology, I would like reviewers to answer the following question (also noted on the manuscript):

You use SEM images to characterize the 3D shape assuming a spherical shape. However, these ash droplets have variable morphology. What is the inaccuracy in size estimation here?

Additionally, it seems you used only 200 grains/1 sample to characterize the ash veneers. Please explain how this is representative. (see Figure 2 for more comments) I would argue this is a rather small amount.

I think the methods would be easily replicable from an observational standpoint.

Suggested Improvements

I don't see any big cause for concern, but I would ask the authors to consider my questions. I appreciate that they've shared their data on a database.

Clarity and Context

The abstract is aptly clear. I note that the order in which implications are introduced in the introduction and conclusion are switched (i.e. TGSD and inner conduit processes), but this is okay as it keeps the hourglass conceptual shape of ideas (broad -> focused -> broad).

References

References are apt. Some consideration could be noted for others who have noticed lapilli amalgamation structures prior to Giachetti et al. (2021) – see line 248 – but this is peripheral.

Your expertise (what you don't bring)

I would suggest someone with regularly works on scaling problems review the paper. My suggestion includes

[REDACTED]

-Dr. K Trafton

Reviewer #2 (Remarks to the Author):

I enjoyed reading your manuscript, below I provide some comments linked by manuscript section. It would have been useful and more time efficient for me to have line numbers in the submission.

Abstract:

Fine and ultra-fine – I suggest defining these sizes.

Introduction:

Last sentences – it is unclear if you are talking about this work or previous work. What is being investigated specifically here?

Results:

Plume height – with reference to what datum?

Figure 1: Remove or improve the map inset. It is unclear in its current form and not that useful. It does not have a panel label, no scale or context. I would suggest that it is removed.

Figure 2: (b) It is hard to see if these are indeed separate particles. Could this not just be an altered/chemically weathered surface? Could it just be pitted due to abrasion -- it is on the edge of a turbulent gas-particle jet as you state. Given this is the main line of evidence in the paper some better, more convincing imagery is needed.

Paragraph starting "We can place..." The link to figure 3 could be better. Where exactly is $St = 1$?

The short-mixed paragraph defining turbophoresis and wall impacts could be confusing, why mix these two processes in such a short paragraph. I would suggest adding the turbophoresis information to the paragraph above and the E_o , O_h , We material to the paragraph below.

Although operating on (potentially) different scales the authors should introduce the previous volcanic droplet impact work done (e.g., "Spatter" Sumner et al., 2005). Related to this, what about droplets that splash and/or rebound on impact?

Figure 3a: It is unclear what the red histogram units/axis labels are. Why is $Re = 1$ marked? This needs a better explanation in the caption and main text. Figure 3b, the regime fields need citations.

Comments around TGSD modification: these comments and references are focussed on silicic systems, the authors conceptual model of post primary fragmentation of droplets still in the molten state is readily observed to modify the (T)GSD of mafic products. References and

statements relating to this would be useful and complement the work.

How is the original particle size distribution created? Is it energetically feasible to create ultra-fine particles at the depths proposed?

Figure 5: it is hard to see what is going on. The images are too small and pixelated.

Reviewer #3 (Remarks to the Author):

In this work, the authors investigated ash particles sintered on fracture surface which was formed during the 2011–2012 explosive eruptions of Cordón Caulle and proposed that the particles sequestered in the shallow subsurface because of turbophoresis and rapid sintering in the conduit after magma fragmentation. Although this is the first theoretical investigation of particle dynamics in the conduit for this eruption, the conceptual model is almost the same as the one previously proposed by the same authors. The absence of a description of volcanic ash particles raises questions about this model.

First of all, petrological studies on the ash particles are needed to confirm that they are sintered magmatic particles. In Fig. 2b, many platy particles are found. Without additional information such as chemical composition, we wonder if these particles are all melt particles. I suspect that some of these particles may be crystals in the magma or precipitated from the gas phase. If so, the sintering model should be revised and the precipitated particles must be removed from data in Fig. 2d.

The authors emphasize that small ash particles were captured and sequestered in the subsurface rather than emitted. However, the amount of emitted small particles is probably not definite, at least not in this study, because the data are not presented. In other words, without comparing the amount of particles captured in the subsurface and those released to the surface, the authors cannot conclude that small ash particles were captured and sequestered in the subsurface rather than emitted.

The authors calculated sintering timescale of melt particles at a temperature of 900°C. The water content is not given in the text but it is likely to be about 0.1–0.3 wt% for a viscosity of $\sim 10^{8}$ Pa s based on the model of Giordano et al (2008). With these parameters, the authors assume that the timescale for relaxation of the melt by surface tension is short enough to allow sintering during interaction with the wall. However, they do not explain why these parameters can be applied to this estimation. In particular, it is assumed to be the magma temperature estimated based on geothermometer (Castro et al., 2013), although no explanation is found in the text. During magma ascent, the temperature does not change? If the temperature drops by only 50°C, the viscosity of the melt increases to $\sim 10^{9}$ Pa s and the relaxation timescale increases by one order of magnitude, i.e., the relaxation timescale by surface tension (~ 550 – 1104) is longer than the interaction timescale (7–647 s).

REVIEWER COMMENTS

Reviewer #1 (Remarks to the Author):

Reviewer Report

Key results

Farquharson and authors propose a model in which submicron-scale ash particles are preferentially removed from the main flow via turbophoresis and subsequently agglomerate on conduit margins. Their model is based on physical field observations of the 2011-2013 eruption of Cordon Caulle and its resulting products, the latter which notably includes ash-veneered obsidian lava – the crux of their observational argument. Like tuffisites, they interpret such veneers represent upper conduit ash venting pathways. Feasibility of such interpretation of ash migration is supported by inner conduit process timescales and scaling arguments.

Validity

The authors propose a model that furthers strengthens arguments for upper conduit separation/amalgamation of pyroclastic products. Their study neatly fits in with others' work on ash and lapilli aggregation. I see no major flaws which should prohibit its publication, but I do have some suggestions worth considering mainly as noted in the commented manuscript.

We appreciate the comments from the reviewer here, and thank them for their useful suggestions.

Significance

This manuscript is of particular importance to the physical volcanology community, but has broader implications. It underscores the necessity of revisiting whether total grain size distribution (TGSD), as well as the erupted componentry, accurately capture inner-conduit processes. Notably, such ash depletion would remove fine material from the fallout TGSD, and as such result in inaccurate inputs for hazard models. Coupled with other product modification (e.g. secondary fragmentation) TGSDs increasingly seem to reflect less and less the state of magma at fragmentation.

An additional implication of their argument – which fits in the Wadsworth-pyroclastic-origin-of-effusive-products arguments – is that we are severely understanding the amount of fine material produced in the conduit, as the subsequent eruption of obsidian is volumetrically significant in context with the fallout phase. These ash veneers represent yet another fine pyroclastic component found in the effusive phase.

Data and Methodology

While I see no glaring issues with the methodology, I would like reviewers to answer the following question (also noted on the manuscript):

You use SEM images to characterize the 3D shape assuming a spherical shape. However, these ash droplets have variable morphology. What is the inaccuracy in size estimation here?

The reviewer raises a valid point; however, the 3D shape is only used illustratively (e.g. Figure d), rather than quantitatively. We now clarify this in the text (please see section “Microscopy and grain size analysis”), also explicitly acknowledging the variability in particle shape:

“We note that volume (and thus, mass) estimates are approximate, due to the variability in particle morphology, which cannot be accurately determined in two dimensions. Mass distributions are used illustratively, and the measured grain diameters are instead used in any further analyses.”

Additionally, it seems you used only 200 grains/1 sample to characterize the ash veneers. Please explain how this is representative. (see Figure 2 for more comments) I would argue this is a rather small amount.

We agree that additional measurements would be beneficial. We have now prepared four more sets of thin-section samples and increased the measurement number from 200 to 1467. We find that the distributions are comparable between samples, and sub-micron size particles are observed in all image fields. We now provide further information in the Methods:

“All particle size data shown in Fig. 2d, h, l, p were obtained at 4,000–4,300× magnification, and all data in Fig. 6d at 3,700–4,300× magnification: annotated SEM images are provided as Supplementary Material. The number and size of particles that can be classified in this way is dependent on image size and resolution; accordingly, in order to ensure the total particle population was appropriately and representatively sampled, we also measured particles using a wider range of magnifications. Decreasing and increasing the magnification to 500× and 10,000×, respectively, did not appreciably affect the fraction of largest or smallest particles observed. We obtain log-normal particle size distributions for each sample, indicating that we are capturing a full distribution. Altogether, we show data from 1467 measured particles.”

Moreover, we have now supplemented Figure 2 with additional SEM imagery of the newly measured samples:

I think the methods would be easily replicable from an observational standpoint.

Suggested Improvements

I don't see any big cause for concern, but I would ask the authors to consider my questions. I appreciate that they've shared their data on a database.

Clarity and Context

The abstract is aptly clear. I note that the order in which implications are introduced in the introduction and conclusion are switched (i.e. TGSD and inner conduit processes),

but this is okay as it keeps the hourglass conceptual shape of ideas (broad -> focused -> broad).

References

References are apt. Some consideration could be noted for others who have noticed lapilli amalgamation structures prior to Giachetti et al. (2021) – see line 248 – but this is peripheral.

Your expertise (what you don't bring)

I would suggest someone with regularly works on scaling problems review the paper. My suggestion includes

[REDACTED]

-Dr. K Trafton

The reviewer also added a series of comments directly to the manuscript, which are reproduced here:

1. I would suggest a transition sentence at the end of the first paragraph.

We have now added: “Key to determining eruptive source parameters is the total grainsize distribution (TGSD) of erupted volcanic ash particles...”

2. In real TGSD, there are secondary size maxima with distance from the plume due to static aggregation of fine material. There's also the issue of secondary fragmentation during deposition... To what degree do these and other process impact TGSD as opposed to turbophoresis?

We appreciate this comment and suggestion to compare the relative importance of these different processes on the TGSD. As we note in the revised text, our treatment of fines sequestration in the conduit represents a previously undocumented mechanism to influence the TGSD, and notably in the context of an eruption cycle, possibly one of the earliest processes that will operate syneruptively, due to the proximity of the turbophoresis zone to the fragmenting and erupting magma. We now highlight this explicitly in the text: “Unlike TGSD modification driven by post-emplacement agglutination—proposed for Hawaiian-style basaltic systems⁵⁵—we anticipate in-conduit turbophoresis and sintering to be one of the earliest operative syneruptive processes due to the colocation of the fragmentation and particle migration mechanisms.”

The relative importance of secondary maxima due to ash accretion (as seen in Mount St. Helens, 1980) and secondary fragmentation during deposition cannot be estimated with our current dataset but we do agree that any attempt to reconcile the TGSD against both conduit

and external/transport related processes will need a proper natural event and deposits that have not suffered from post-eruption modification to the TGSD (owing to the time since its 2011 eruption, we cannot guarantee that these criteria at Cordón Caulle are met). We also refer the reviewer to text acknowledging secondary processes:

“Other processes—including shear or abrasion mechanisms^{55,56} and related in-conduit comminution processes⁵⁷—act to divorce grain size distributions from the magmatic source conditions, increasing the fines proportion relative to the size distribution produced at true magmatic fragmentation at depth.”

3. You say 2011-2012 here, but the first sentence says 2011-2013. The latter instance has been corrected to “2011–2012.”

4. How do you know this? Based on work by Gardner and Watkins with ash sintering/obsidian pyroclast formation, ash could amalgamate, be ripped up and incorporated into the main flow, and subsequently redeposited farther up conduit. The degree of volatile (H₂O/CO₂) and textural heterogeneity of sintered ash suggests a complex history of amalgamation wherein individual mostly-quenched ash pieces mostly quenched at different depths come together... And these amalgamates could re-amalgamate later on.

This is an interesting point. Based on SEM analysis, we do not observe evidence for multiple stages of sintering in our samples, we acknowledge that such a process could be feasible. However, we note that the majority of the discrete particles are so small (i.e. submicron size) that it seems unlikely that they are amalgamated products of even finer fragmentation. This is in contrast to the centimetric chips described by Gardner and co-authors.

5. If the conduit is less than cylindrical, more of a tuffisitic network, or even a dike system, how would this impact the flow in the conduit and thus your interpretation?

The model is for a general wall-bounded flow, thus consistent with an idealised tuffisite, vent, or conduit geometry. We agree that, in nature, the architecture of ash transport pathways is undoubtedly more complex; however, without detailed constraints of the true geometry, there is little tangible benefit to arbitrarily increasing the complexity of the model. One key parameter that could be affected is the relative velocity of particles: increased channel tortuosity would presumably influence the cross-channel velocity differential and provide additional capture points for particles. The fundamental operative mechanisms would still be the same, however: clustering, decoupling, turbophoresis, sintering.

6. Does the shape of the particles control in part how likely they are to amalgamate as well? Certain shapes I could see being conducive to separation from the main flow and thus

aggregation. For instance, there may be a particle sufficiently small to be affected by turbophoresis, but if it is elongate in nature, wouldn't it be more likely to "go with the flow"?

This is a great point by the reviewer. The effects of particle shape on turbophoresis is an emerging field, and no studies—to the authors' knowledge—have been done on natural systems. However, preliminary results from large eddy simulations (e.g. Njobuenwu and Fairweather 2014) indicates that, under certain idealised conditions, non-spherical particles show a lower deposition rate at the walls of a channel (a proxy for the incidence of turbophoresis) relative to spherical particles of equal equivalent volume diameter. On the other hand, point-particle modelling by Yuan et al. (2017) indicates that in high-inertia systems, the effect of particle shape on translational motion is marginal, if extant. This is also echoed by Arcen et al. (2017), who highlights that shape has a large effect on particle velocity and rotational dynamics, but comparatively little effect on translational dynamics (i.e. movement from the centreline to the wall).

Njobuenwu, D.O. and Fairweather, M., 2014. Large eddy simulation of non-spherical particle deposition in a vertical turbulent channel flow. In *Computer Aided Chemical Engineering* (Vol. 33, pp. 907-912). Elsevier.

Yuan, W., Andersson, H.I., Zhao, L., Challabotla, N.R. and Deng, J., 2017. Dynamics of disk-like particles in turbulent vertical channel flow. *International Journal of Multiphase Flow*, 96, pp.86-100.

Arcen, B., Ouchene, R., Khalij, M. and Tanière, A., 2017. Prolate spheroidal particles' behavior in a vertical wall-bounded turbulent flow. *Physics of Fluids*, 29(9), p.093301.

We now mention this explicitly in the text:

“We observe a diversity of particle shapes in our samples (Fig. 2). However, we do not account for this explicitly, as recent research suggests that while particle shape has a large effect on particle velocity and rotational dynamics, it has comparatively little effect on translational dynamics (i.e. movement from the centreline to the wall)³⁵.”

A deeper interrogation of the interaction of high-temperature, non-spherical melt particles in a turbulent field would presumably necessitate a combination of experimental work and physics-based direct numerical simulations—an exciting avenue for future research.

7. I appreciate you addressing all of these processes/characteristics that could impact particle organization. I'm curious how conduit shape affects turbophoresis (there's another comment somewhere about this).

Please see response to comment 5, above.

8. Yes, important to note.

We thank the reviewer.

9. Can you also compare ash particle shape as a proxy for degree of energetics of fragmentation? That is to say, in addition to the size of the particles being controlled by the intensity of the process, could the shape of proto ash particles be as well? I see your Figure 2, and the different shapes of the ash, and think it would be helpful to see them in context with ash formed by other processes. It would really emphasize your argument and provide a schema to get all readers on board regardless of their background. A conceptual size-shape diagram in the supplemental could be useful.

We observe a wide variety of particle shapes, now better illustrated in the revised Figure 2. In particular, we observe ash fragments preserved after having undergone different amounts of sintering. Implicitly, the more angular fragments represent the shape immediately following fragmentation, whereas the more spherical or fluidal particles represent more evolved sintering. As well as Figure 2, we show examples in Figure 6, and discuss this further in the attendant text, now appearing in the section “In vent observations from the 2011-2012 Cordón Caulle eruption”:

“Four samples were analysed using scanning electron microscopy: AN1, AN2, CCTVAIP, and CCVP. Imagery of ash veneers reveals a diversity of particle morphologies, including vesicle-free, glassy ash shards and rounded fragments with grain diameters in the range 0.1–100 μm (Fig. 2a–c, e–g, i–k, m–o) and predominantly $<1 \mu\text{m}$ (Fig 2d, h, l, p). Ash morphologies range from angular (e.g. Fig. 2b, f, g, m), with conchoidal fracture surfaces, to near-spherical (e.g. Fig. 2e, j) and droplet-like with narrow necks between ash particles and other surfaces (Fig. 2c)...”

“...As well as necking (Fig. 2c), other evidence points to the veneers being composed of a continuum of variably sintered particles adhered to a competent substrate: in Fig. 2b and 2e, for example, discrete particle shapes (platy and near-spherical, respectively) can be clearly observed, suggesting the earliest stages of contact sintering. In Fig. 2k, adjacent particles exhibit different stages of welding onto a larger grain. In Fig. 2o, the tumulus-like lumps (highlighted) represent the advanced stages of droplet sintering to a planar substrate.”

“...Notably, sample CC1960 exhibits clear sintering textures, including a transition from densely packed partially welded samples (both spherical and angular: Fig. 6c) to a fully cohesive sintered surface (Fig. 6a,b).”

10. An interesting word choice, but I agree.

We thank the reviewer.

11. Worth noting that this recognition was also found by others years before Trafton and Giachetti (2021) EPSL and Giachetti et al. (2021) GEOLOGY papers... Check references in Giachetti et al. (2021).

We thank the reviewer for highlighting these other references.

12. How accurate do you think your grain diameter measurements are in capturing the 3D volume, especially when you see grains with variable morphology? Is there any way to calculate % error in measurement? How were these slides prepared? I can imagine that depending on how you slice your rock for thin section, you would have different estimates of the particle geometry.

The reviewer makes a good point here, in that the majority of the samples are not 100% spherical. We now explicitly mention in the Methods that we measure the “semimajor grain axis from representative 2D SEM images...” Moreover, we now state “As we use the semimajor axis, particle diameters are nominally maxima, notwithstanding their extension in 3D.” We note, however, that irrespective of measurement uncertainties and the pitfalls of assuming three-dimensional geometries from two-dimensional imagery, the measured distributions are several orders of magnitude smaller than the mean grain sizes assumed or estimated using other methods, which is the key element of this article.

13. How ubiquitous are these surfaces on the lava? Is there a marked relationship with the frequency of these features and their location to the vent?

Similar features can be observed all throughout the flowfield. It is probable that these are relicts of near-vent gas transport channels which were then rafted downstream as the lava flowfield continued to evolve. Hence, the more distal examples (now shown and interpreted in greater detail in the revised Figure 6, below), are the oldest.

[REDACTED]

Figure 6 | Extended field observations. **a–c** SEM images of sample CC1960, collected from Cordón Caulle 1960 CE eruptive vent. In **c**, both near-spherical and angular particles can be observed. **d** Particle size distribution for sample CC1960, as in Figure 2. **e** Detail of veneered fracture surface distal from the main vent at Cordón Caulle. Arrows highlight millimetric-to-centimetric clasts sintered to the fracture surface. Sharpie Marker pen is ~14 cm. **f** Schematic interpretation of **e**, indicating laterally juxtaposed zones that had experienced various primary operative processes during emplacement (turbophoresis, sintering, crystallisation/textural maturation). Note that a planar fracture conduit geometry facilitates this ordering of textural zones, resulting in a local time-transgressive sequence of lava production. **g** Detail of a veneered fracture surface at Little Glass Mountain (Medicine Lake volcano, northern California). Reddish ash-coated surface is highlighted by the arrows. Inset shows ash-coated surface in context of exposed fracture plane.

14. Wadsworth et al (2020) argued for the explosive origin of lava. So then, do your obsidian samples with the veneer also have ash grain boundaries in the black portion? Or have the grain boundaries been eliminated? If so, why is it that the veneers were preserved on the red part while those in the black matrix sintered completely?

This is a perceptive question. As highlighted in Wadsworth et al. [2020], “[t]he end result of sintering of crystal-poor rhyolite volcanic ash particles is a dense melt, nearly indistinguishable from magma that was never fragmented, except for the low water

content.” For the most part, individual clast boundaries will have been eliminated due to sintering-driven destruction of inter-clast porosity. However, in some cases, relict grain boundaries have been preserved in obsidian chips—

[REDACTED]

While in the context of the current study, we did not examine the coherent material in detail (the focus instead being on the oxidised particles), the images above do hint at the possibility that ash grain boundaries are preserved elsewhere in the lava (i.e. in the black portion). The primary difference between the red veneers studied here and black fracture planes elsewhere in the system is presumably due to oxidation resulting from increased air mixing. (This could be a function of their being located near the surface during waning eruptive periods.) Notably, increased oxidation means that Fe_2O_3 is enhanced relative to FeO , causing higher viscosity and slower sintering times compared to the black, non-oxidised counterparts. This would explain why the red veneers are seemingly better preserved. However, more detailed microanalysis of tuffisites and other ash-venting structures from around the flowfield may yet reveal comparable preserved structures in black obsidian. In the revised Figure 6 (two panels shown below), we highlight and interpret an ash-venting structure from the distal part of the flowfield. In this case we can see a gradation from the oxidised central part of the ash-venting pathway—which would have been open and subject to turbophoresis—to densely sintered selvages, and finally to the mature lava on either side. Further research would be required to determine the extent to which the fracture walls have been built “from the inside out” as sintering progresses.

[REDACTED]

We now speculate as such in the text:

“Figure 6e (interpreted in Fig. 6f) illustrates that the walls of actively venting fractures are the sites of particle capture, sintering, and ultimate densification of ash into coherent lava. This process may be efficient and operate over quick enough timescales to create zones of accretionary lava at the margins of vents (Fig. 6e,f).”

15. Did you only do this for one image from one sample? How did you choose which image to use? How is this representative of the greater sample and eruption processes at large?

We agree with the reviewer on this point: please refer to the response above to comment on Data and Methodology. We have now increased our sample count from 200 to 1467, and have included images and data from four additional samples.

16. I notice you have a 25m scale for conduit size. How did you determine conduit size, and how does your interpretation change with conduit size? For instance, does the horizontal profile of where these processes are occurring change based on conduit size? Would you then expect the fine/coarse particle distribution to change as a function of conduit size?

The scale on the figure is based on the crater dimensions observed during and after the eruption (see Figure below). However, this is an order-of-magnitude approximation: in reality, the crater diameter and morphology varied substantially throughout the course of the eruption, and it is logical to assume that the conduit diameter varied similarly. We now mention this explicitly:

“Scales are approximate, as the crater—and presumably the conduit—diameter and morphology changed throughout the course of the eruption.”

The total conduit size is less important in the context of our observations, as we are focussed on the relatively narrower ash-venting pathways fracturing the (re-)amalgamated magma within the upper part of the conduit (see Figure 5 in the revised manuscript)—specifically a 0.01 m wall-bounded flow in the case of (revised) Figure 4. Nevertheless, the processes described are self-similar across spatial scales, as long as flow remains turbulent. Thus, a larger wall-bounded geometry (such as a large tuffsite or the main conduit), would yield a region in $\langle u \rangle - R$ space that corresponds to a different particle size distribution within the $St = 1$ bounds (Figure 4). With sufficient knowledge of the geometry of a venting pathway of interest—whether the main conduit or a subsidiary fracture—the particle size distribution subject to capture via turbophoresis and sintering could be calculated using the analytical framework described in this study.

17. Formatting note - Do you mean to have the text wrapping this way? It's also a problem in Figure 4.

Thanks for highlighting this: this has been corrected.

Reviewer #2 (Remarks to the Author):

I enjoyed reading your manuscript, below I provide some comments linked by manuscript section. It would have been useful and more time efficient for me to have

line numbers in the submission.

We appreciate the reviewer's comment that they enjoyed reading the manuscript, and thank them for their constructive comments.

Abstract:

Fine and ultra-fine – I suggest defining these sizes.

We now state:

“...demonstrating that fine (<63 μm diameter) and ultra-fine (<2.5 μm diameter) ash particles are captured and sintered to fracture surfaces...”

Introduction:

Last sentences – it is unclear if you are talking about this work or previous work. What is being investigated specifically here?

We have now added:

“In this study, we present microtextural data from within the Cordón Caulle vent, and build a conceptual model for the capture of an ultra-fine ash fraction within the shallow vent architecture. We further highlight implications of the fact that the emitted products of explosive fragmentation may be fundamentally decoupled from the eruption source parameters during silicic eruptions.”

Results:

Plume height – with reference to what datum?

Above sea level. We have now added this to the figure caption.

Figure 1: Remove or improve the map inset. It is unclear in its current form and not that useful. It does not have a panel label, no scale or context. I would suggest that it is removed.

We now include two georeferenced satellite images and an inset globe in order to situate our study area:

Editorial Note: Satellite data courtesy of NASA's Earth Observatory
<https://earthobservatory.nasa.gov/>.

Figure 2: (b) It is hard to see if these are indeed separate particles. Could this not just be an altered/chemically weathered surface? Could it just be pitted due to abrasion -- it is on the edge of a turbulent gas-particle jet as you state. Given this is the main line of evidence in the paper some better, more convincing imagery is needed.

We appreciate that these particles may not necessarily appear discrete (a consequence of their variably sintered character). We have now obtained additional imagery from multiple samples to highlight their morphology, alongside geochemical data as evidence that this is not a chemically altered surface. Notably, no geochemical evidence of precipitates or weathering were detecting using point EDX analyses, and sample element maps highlight the relative compositional homogeneity of the particles and the silicate substrate beneath.

The text has been updated accordingly:

“As well as necking (Fig. 2c), other evidence points to the veneers being composed of a continuum of variably sintered particles adhered to a competent substrate: in Fig. 2b and 2e, for example, discrete particle shapes (platy and near-spherical, respectively) can be clearly observed, suggesting the earliest stages of contact sintering. In Fig. 2k, adjacent particles exhibit different stages of welding onto a larger grain. In Fig. 2o, the tumulus-like

lumps (highlighted) represent the advanced stages of droplet sintering to a planar substrate.

Despite the variety of microtextures observed across the four samples (Fig. 2), including angular fragments such as highlighted in Fig. 2m, energy dispersive X-ray (EDX) analysis indicates that the veneers are broadly homogeneous in composition, revealed by elemental mapping (Fig. 3a–c) and point analyses (Fig. d–h). Angular fragments (Fig. 3d, f) are largely indistinguishable from more rounded and fluidal particles (Fig. 3g,h); the substrate (Fig. 3d,e) appears relatively depleted in mobile cations (Na, Al, K) relative to the variably sintered particles. The lack of additional elements such as S indicates that the particles are rhyolitic glass, with a minor plagioclase component consistent with phenocryst populations, but with no evidence of mineral precipitation. Additional EDX data are provided as Supplementary Material.”

Figure 2 | Particle sizes in sintered veneers on in-vent lava fractures that fed ash-venting during hybrid explosive-effusive eruptions.

Figure 3 | Geochemistry of the sintered veneers. **a, b** Energy-dispersive X-ray spectroscopy (EDX) element maps of sample CCTVAIP, with Si, O, and additional elements overlain on an SEM image. **c** As **a**, for sample CCVP. **d** Point EDX analyses for sample CCVP, with spectra for points s1–s4 (the substrate) shown in **e**, and points s5–s10 (discrete particles) in **f**. Peaks for C, O, Na, Al, Si, and K are highlighted. Note white box showing area of **Fig. 2m**. **g** As **d**, for sample CCTVAIP. Spectra for points s11–s14 are shown in **h**.

Paragraph starting “We can place...” The link to figure 3 could be better. Where exactly is $St = 1$?

Given the range of potential eddy sizes (from the Kolmogorov scale to the “outerscale,” i.e. the fracture width), there is no single point in $\langle u \rangle - R$ space where $St = 1$. Rather, St can equal 1 anywhere between the dashed and solid lines on Figure 3, depending on eddy size. Outwith these lines, St cannot equal 1. We now state this more explicitly for the sake of clarity:

“...measured in the vent at Cordón Caulle. **The unshaded region indicates the range of $\langle u \rangle$ and R where it is possible for St to equal 1 (dependent on eddy size): outside this region, St must be lower than unity (below the $St = \lambda_p/\lambda_K$ line) or greater than unity (above the $St = \lambda_p/\lambda_o$ line). Notably...**”

The short-mixed paragraph defining turbophoresis and wall impacts could be confusing, why mix these two processes in such a short paragraph. I would suggest adding the turbophoresis information to the paragraph above and the E_o , Oh , We material to the paragraph below.

We have changed this as suggested.

Although operating on (potentially) different scales the authors should introduce the previous volcanic droplet impact work done (e.g., “Spatter” Sumner et al., 2005). Related to this, what about droplets that splash and/or rebound on impact?

This is a good suggestion. We now cite Sumner et al. (2005). The regime plot (Figure 4c) is one in which splashing effects could be delineated (e.g. splashing is found to be prominent in regime I, which would involve substantially higher velocities (and therefore Weber numbers) than are reasonable for Cordón Caulle eruptions or silicic eruptions in general. By contrast, at lower Ohnesorge number, regime I is met at eruption velocities, and such lower Ohnesorge numbers would be typical of basaltic eruptions that produce spatter (as investigated by Sumner et al. 2005). Therefore, while we make reference to this in the new submitted version, we conclude that it is not relevant for the Cordón Caulle case. New text reads:

“We note that droplet splashing can occur in regime I, and is typical of low viscosity droplets such as produced at basaltic eruption conditions (cf. Sumner et al. 2005), and/or very high impact velocities.”

Note that in all experimental investigations under isothermal magmatic conditions of melt impact at the particle/droplet scale (e.g. Schiaffino and Sonin 1997; Giehl et al. 2017), it has been observed that droplets will stick on impact.

Figure 3a: It is unclear what the red histogram units/axis labels are.

This figure has been re-drawn with an additional y-axis for clarity.

Why is $Re = 1$ marked? This needs a better explanation in the caption and main text.

Originally, $Re=1$ was included here as it represents the velocity below which the fluid cannot be considered turbulent. For clarity, this threshold has been removed.

Figure 3b, the regime fields need citations.

The appropriate reference to Schiaffino and Sonin (1997) is included in the figure caption.

Comments around TGSD modification: these comments and references are focussed on silicic systems, the authors conceptual model of post primary fragmentation of droplets still in the molten state is readily observed to modify the (T)GSD of mafic products. References and statements relating to this would be useful and complement the work.

We appreciate this comment from the reviewer. We now refer to Parfitt (1988), who highlight relative coarsening of near-vent clasts due to post-eruptive clast agglutination.

Parfitt, E.A., 1998. A study of clast size distribution, ash deposition and fragmentation in a Hawaiian-style volcanic eruption. *Journal of Volcanology and Geothermal Research*, 84(3-4), pp.197-208.

The text now reads:

“Unlike TGSD modification driven by post-emplacement agglutination—proposed for Hawaiian-style basaltic systems⁵⁵—we anticipate in-conduit turbophoresis and sintering to be one of the earliest operative syneruptive processes due to the collocation of the fragmentation and particle migration mechanisms.”

How is the original particle size distribution created? Is it energetically feasible to create ultra-fine particles at the depths proposed?

Given that the particles exist, there must be a physical mechanism by which to generate them. However, existing empirical relations between particle size and fragmentation efficiency generally do not consider particles sizes of the order of magnitudes presented here. The exponential nature of hydrodynamic fragmentation models (e.g. Zimanowski et al. 2003) means that extrapolating to submicron scales yields implausibly high fragmentation energies (interfacial acceleration rates of $>10^7$ m s⁻²). Thus, primary generation of fine and ultra-fine ash must be the result of a brittle fragmentation mechanism, for which there is no generally accepted theory. It is our hope that future generations of fragmentation models account explicitly for the potential generation of submicron-scale particles.

We have now added this to the text, in the section “Grain size fractionation”:

“The exponential nature of hydrodynamic fragmentation models means that extrapolating to submicron scales yields implausibly high fragmentation energies. Primary generation of fine and ultra-fine ash must therefore be the result of a brittle fragmentation mechanism, for which there is no generally accepted theory⁴⁶. It is our hope that future generations of fragmentation models account explicitly for the potential generation of submicron-scale particles.”

Figure 5: it is hard to see what is going on. The images are too small and pixelated.

This figure has been revised so that more detail is visible:

[REDACTED]

Figure 6 | Extended field observations. **a–c** SEM images of sample CC1960, collected from Cordón Caulle 1960 CE eruptive vent. In **c**, both near-spherical and angular particles can be observed. **d** Particle size distribution for sample CC1960, as in Figure 2. **e** Detail of veneered fracture surface distal from the main vent at Cordón Caulle. Arrows highlight millimetric-to-centimetric clasts sintered to the fracture surface. Sharpie Marker pen is ~14 cm. **f** Schematic interpretation of **e**, indicating laterally juxtaposed zones that had experienced various primary operative processes during emplacement (turbophoresis, sintering, crystallisation/textural maturation). Note that a planar fracture conduit geometry facilitates this ordering of textural zones, resulting in a local time-transgressive sequence of lava production. **g** Detail of a veneered fracture surface at Little Glass Mountain (Medicine Lake volcano, northern California). Reddish ash-coated surface is highlighted by the arrows. Inset shows ash-coated surface in context of exposed fracture plane.

Reviewer #3 (Remarks to the Author):

In this work, the authors investigated ash particles sintered on fracture surface which was formed during the 2011–2012 explosive eruptions of Cordón Caulle and proposed that the particles sequestered in the shallow subsurface because of turbophoresis and rapid sintering in the conduit after magma fragmentation. Although this is the first

theoretical investigation of particle dynamics in the conduit for this eruption, the conceptual model is almost the same as the one previously proposed by the same authors. The absence of a description of volcanic ash particles raises questions about this model.

First of all, petrological studies on the ash particles are needed to confirm that they are sintered magmatic particles. In Fig. 2b, many platy particles are found. Without additional information such as chemical composition, we wonder if these particles are all melt particles. I suspect that some of these particles may be crystals in the magma or precipitated from the gas phase. If so, the sintering model should be revised and the precipitated particles must be removed from data in Fig. 2d.

We thank the author for these comments. We have now performed additional analysis, including additional microscopy and compositional analysis. We can confirm that the particles are composed of rhyolitic glass (both spot analyses and EDX maps confirm this). This geochemical data is now provided in a new figure, reproduced here:

Figure 3 | Geochemistry of the sintered veneers. **a, b** Energy-dispersive X-ray spectroscopy (EDX) element maps of sample CCTVAIP, with Si, O, and additional elements overlain on an SEM image. **c** As **a**, for sample CCVP. **d** Point EDX analyses for sample CCVP, with spectra for points s1–s4 (the substrate) shown in **e**, and points s5–s10 (discrete particles) in **f**. Peaks for C, O, Na, Al, Si, and K are highlighted. **g** As **d**, for sample CCTVAIP. Spectra for points s11–s14 are shown in **h**.

Extended EDX results are also provided as a Supplementary File. Additional textural and compositional description has been added to the maintext:

“As well as necking (Fig. 2c)—a clear indicator of partial sintering—other evidence points to the veneers being composed of variably sintered particles adhered to a competent substrate: in Fig. 2b and 2e, for example, discrete particle shapes (platy and near-spherical, respectively) can be clearly observed, suggesting the earliest stages of contact sintering. In Fig. 2k, adjacent particles exhibit different stages of welding onto a larger grain. In Fig. 2o, the tumulus-like lumps (highlighted) represent the advanced stages of droplet sintering to a flat substrate.

Despite the variety of microtextures observed across the four samples (Fig. 2), including angular fragments such as highlighted in Fig. 2m, energy dispersive X-ray (EDX) analysis indicates that the veneers are broadly homogeneous in composition, revealed by elemental mapping (Fig. 3a–c) and point analyses (Fig. d–h). Angular fragments (Fig. 3d, f) are largely indistinguishable from more rounded and fluidal particles (Fig. 3g,h); the substrate (Fig. 3d,e) appears relatively depleted in mobile cations (Na, Al, K) relative to the variably sintered particles. The lack of additional elements such as S (which would be manifest as a peak in counts ~ 2.3 keV) indicates that the particles are rhyolitic glass, with no evidence of mineral precipitation. Additional EDX data are provided as Supplementary Material.”

We highlight that the platy and (near-)spherical particles observed in the analysed samples are largely indistinguishable from a compositional perspective (compare spectra related to panels d and g above). None of our analyses indicate precipitation processes (there was no evidence of Sulphur or Chlorine, which could indicate precipitates of NaCl or CaSO₄, for example), and only minor peaks in Al, Na, and K were detected (consistent with a rhyolitic glass or trace fragments of plagioclase).

The authors emphasize that small ash particles were captured and sequestered in the subsurface rather than emitted. However, the amount of emitted small particles is probably not definite, at least not in this study, because the data are not presented. In other words, without comparing the amount of particles captured in the subsurface and those released to the surface, the authors cannot conclude that small ash particles were captured and sequestered in the subsurface rather than emitted.

Evidence of particle capture is manifest in the presence of ultra-fine ash veneers that are the focus of this study. Evidence of the *absence* of this ash fraction in emitted products is a little harder to demonstrate. However, in Figure 5a (shown below), we compare the size distribution measured in our samples to size distribution reconstructions from Reckziegel et al. [2019] and Costa et al. [2016]. In particular the Costa et al. [2016] reflects the best estimate of total emitted grain-size distributions derived from field data, based on ash sampling. Although Costa et al. note a fine tail in the distribution of Cordón Caulle ash, there is very little overlap between their data and the data presented in our study. In particular,

the sub-micron fraction is not reflected in the field data, despite representing a significant proportion of ash particles measured in our study. The existence of this grain size fraction in the near-surface ash vents, coupled with the lack of this ash fraction reported in field deposits, indicates that ultrafine ash particles are generated within the conduit but not emitted at the surface.

The figure has been replotted for additional clarity.

The caption has also been updated accordingly:

“Compiled data in this study reflect a captured in-conduit fine ash fraction, characterised by a mean (as-measured) diameter d of 9.94×10^{-7} m. For each dataset, best-fit lognormal curves have been overlain. Distribution assumed by Reckziegel et al. [ref. 29] are approximately unimodal and described by mean of $d = 3.10 \times 10^{-4}$ m (based on lognormal assumption). Data of Costa et al. [ref. 28], reconstructed from field data, are bimodal, described by lognormal peaks at $d = 1.34 \times 10^{-5}$ and $d = 5.39 \times 10^{-3}$ m. Mean and $\pm 2\sigma$ range are highlighted for data of this study and ref. 29; these values are shown for each of the peaks of ref. 28. Note different bin size between panels.”

The authors calculated sintering timescale of melt particles at a temperature of 900°C. The water content is not given in the text but it is likely to be about 0.1–0.3 wt% for a viscosity of $\sim 10^8$ Pa s based on the model of Giordano et al (2008). With these parameters, the authors assume that the timescale for relaxation of the melt by surface tension is short enough to allow sintering during interaction with the wall. However, they do not explain why these parameters can be applied to this estimation. In particular, it is assumed to be the magma temperature estimated based on geothermometer (Castro et al., 2013), although no explanation is found in the text. During magma ascent, the temperature does not change? If the temperature drops by only 50°C, the viscosity of the melt increases to $\sim 10^9$ Pa s and the relaxation timescale increases by one order of

magnitude, i.e., the relaxation timescale by surface tension ($\sim 550\text{--}1104$) is longer than the interaction timescale ($7\text{--}647$ s).

First, we note that both of the timescales we compare—the droplet-wall interaction timescale λ_d and the sintering timescale λ_r —are dependent on the droplet viscosity and that this dependence is linear in both cases. This implies that if the droplet were to cool during transport (and be cooler than magmatic temperature at the conditions of droplet impact), then both timescales would be affected by the *same* amount. In turn, this implies that the sintering timescale and the interaction timescale will remain comparable, even in the case of a cooling droplet.

Second, although it is compelling if droplet the sintering timescale is generally less than the interaction timescale, this is not a requirement for sticking. All existing work on droplet interactions with surfaces that have used silicic droplets show that if the droplet hits the surface in the molten state, then it will stick (e.g. Giehl et al. 2018; Pearson & Brooker 2020). Whether or not the droplets will sinter to form a densified deposit—inferred here to be required to produce lava—is then associated with the sintering timescale as droplets accumulate on the surface in layers. Therefore, we conclude that while cooling probably did occur during transport, it was not sufficient to drop the droplet temperatures below the glass transition, and that therefore the droplets were molten (inferred on the basis of observed sticking). The surfaces are variably sintered (e.g. Figure 2), suggesting that after droplet interaction and sticking, the surfaces were held for variable times up to the sintering timescale (e.g. to produce the smooth sintered surfaces in Fig. 2j, Fig. 2m, etc.). Other surfaces, meanwhile, did not experience times as long as the sintering time (e.g. to produce Fig. 2b). This could be associated with temperature heterogeneities, or different pathways being utilised at different times, or both.

In acknowledgment of these processes, and in response to the Reviewer comment, we now add the following text:

“It is compelling that the sintering timescale is generally less than the droplet-wall interaction timescale (i.e. $\lambda_r \leq \lambda_d$); however, we highlight that this is not a strictly necessary condition for droplets to stick to the wall in this system: molten silicic droplets will, in all likelihood, stick when interacting with a hot surface^{42,44}. This means that λ_d is therefore indicative of the initial stick and spread dynamics. As more droplets stick and accumulate a surface deposit, the sintering time λ_r becomes more relevant, revealing the most conservative time required for the deposit to densify to a non-porous state. In Figure 2, it is clear that particles have variably undergone full sintering (cf. Fig. 2b with Fig. 2j), and that where individual particles can be seen, they have only variably spread onto the substrate they adhere to. This is consistent with our finding that λ_r and λ_d are predicted to be of a similar order of magnitude, and implies that different regions of these surfaces are

likely to be at different temperatures, accounting for the observed variability in texture. Both λ_r and λ_d have a linear dependence on the droplet viscosity, such that any syn-eruptive cooling during transport that would serve to increase these timescales, would do so proportionally to both.”

In terms of the chosen temperature value, Castro et al. (2013) use Fe–Ti oxide mineral geothermometry to determine the magma storage temperature of Cordón Caulle as ~870–920 °C. Evidence suggests that the eruption temperature likely remains near 900 °C, as such a temperature is required in order to reconcile well-calibrated viscosity models with lava flows that were exceptionally mobile, even years after emplacement (Tuffen et al. 2013), once the effects of the crystal cargo is accounted for (Farquharson et al. 2015). As such, this implies relatively little heat loss during magma ascent in the conduit.

Castro, J.M., Schipper, C.I., Mueller, S.P., Militzer, A.S., Amigo, A., Parejas, C.S. and Jacob, D., 2013. Storage and eruption of near-liquidus rhyolite magma at Cordón Caulle, Chile. *Bulletin of Volcanology*, 75(4), pp.1-17.

Tuffen, H., James, M.R., Castro, J.M. and Schipper, C.I., 2013. Exceptional mobility of an advancing rhyolitic obsidian flow at Cordón Caulle volcano in Chile. *Nature communications*, 4(1), pp.1-7.

Farquharson, J.I., James, M.R. and Tuffen, H., 2015. Examining rhyolite lava flow dynamics through photo-based 3D reconstructions of the 2011–2012 lava flowfield at Cordón-Caulle, Chile. *Journal of Volcanology and Geothermal Research*, 304, pp.336-348.

We now justify our choice of temperature in the text: “...where $T = 900$ °C is assumed to be a representative magmatic temperature based on geothermometry³⁹.”